# IQA-Spider: Unifying Multi-Granularity Image Quality Assessment with Reasoning, Grounding and Referring

**Xinge Peng** [1] [*]  **Yiting Lu** [1] [*]  **Xin Li** [1]  **Zhibo Chen** [1]

## Abstract

We present IQA-Spider, the first image quality assessment (IQA) framework that unifies reasoning, grounding, and referring into a single LMM-based framework for multi-granularity quality understanding. Existing LMM-based IQA methods typically support only partial perception dimensions, such as quality description and question answering (*i.e.*, reasoning) or pixel-level grounding. This limitation largely stems from the absence of (i) a unified task and data formulation and (ii) effective optimization paradigms for multi-granularity learning. To address these limitations, we formulate a rigorous four-task paradigm covering global and local quality description, pixel-level grounding, and region-level referring. Based on this formulation, we construct a corresponding IQA dataset with a scalable and automatic annotation pipeline, thereby providing a solid foundation for unified multi-granularity learning. To further enable unified perception, we adopt a conflict-free two-stage design that progressively extends text-level multi-granularity understanding to pixel-level grounding: (i) the first stage equips the model with fine-grained text-level reasoning across multiple IQA tasks, and (ii) the second stage introduces a training-free text-to-point grounding paradigm, which bridges textual semantics and pixel-level perception by mapping token logits to spatial coordinates. Based on these efforts, we achieve IQA-Spider with unified multi-granularity explainable image quality assessment. Extensive experiments across multiple benchmarks demonstrate strong performance, validating the effectiveness and versatility of the proposed formulation and framework. Our code and dataset will be released at:

[*]Equal contribution  [1] University of Science and Technology of China, Hefei, China. Correspondence to: Xin Li <xin.li@ustc.edu.cn>, Zhibo Chen <chenzhibo@ustc.edu.cn>.

*Proceedings of the $43^{rd}$ International Conference on Machine Learning*, Seoul, South Korea. PMLR 306, 2026. Copyright 2026 by the author(s).

https://github.com/Helen1p/IQA-Spider.git.

## 1. Introduction

As a fundamental task in visual signal processing, image quality assessment (IQA) aims to evaluate perceptual quality by modeling the human visual system (HVS). With the rapid growth of streaming services and the explosive increase of user-generated visual content, IQA has evolved from a standalone quality scoring problem into a practical requirement for explainable quality understanding, which is critical for guiding image compression, image restoration, and preference optimization for AIGC content, ultimately improving user experience.

Recently, large multi-modal models (LMMs) have significantly advanced IQA by enhancing both scoring accuracy and natural-language explainability. Early studies such as Q-Bench (Wu et al., 2023a) demonstrate that LMMs possess non-trivial perceptual quality awareness, while Q-Align (Wu et al., 2023b) further validates their potential for visual quality scoring. Subsequent works extend these capabilities to quality description and reasoning (Wu et al., 2024a; You et al., 2025b; Lu et al., 2025b), enabling more human-interpretable assessments. However, existing LMM-based IQA methods largely remain global and coarse-grained. As observed in DepictQA-Wild (You et al., 2025b), the generated descriptions often fail to localize distortions or identify region-specific quality issues, revealing a critical limitation in fine-grained quality perception. Although Q-Ground (Chen et al., 2024a) introduces distortion-type grounding, its formulation is tightly coupled to a narrow distortion taxonomy and does not generalize well to broader quality understanding tasks.

While many recent efforts (Xing et al., 2025; Embodied) in IQA primarily focus on scaling up data volume or expanding to new domains and scenarios, such strategies alone do not fully address the missing capability of region-aware and fine-grained quality understanding. We do not aim to introduce a very large-scale dataset as an end goal; instead, we present a systematic and extensible task-and-data formulation for multi-granularity IQA, extending the scope of image quality understanding from global-level to region-level. To support

this reformulation, we first formalize multi-granularity IQA as a unified family of complementary tasks, rather than a single global objective. Specifically, we define four task paradigms that capture different aspects of quality understanding: *(1) Global Quality Description*, *(2) Local Quality Description*, *(3) Visual Quality Grounding*, and *(4) Visual Quality Referring*. These tasks jointly enable global reasoning, region-level reasoning, and pixel-level quality analysis, providing a structured basis for developing explainable IQA systems.

Based on this task formulation, we introduce IQA-Spider-33K, a multi-granularity dataset that instantiates the proposed tasks in a practical and scalable manner. Although moderate in scale, the dataset is designed with a clear focus on principled construction rather than sheer data volume. Compared to existing explainable IQA datasets (Wu et al., 2024a; Chen et al., 2024a;c), our dataset design offers three key advantages. **(i) Comprehensive Task Taxonomy and Degradation Space.** We decompose region-aware grounding and referring into structured sub-tasks to accommodate diverse application scenarios. Moreover, we manually define a comprehensive degradation space covering both synthetic and in-the-wild distortions, ensuring broad and realistic distortion coverage. **(ii) Efficient and Scalable Annotation Pipeline.** We propose a hybrid annotation pipeline that leverages SAM-based tools and open-source LMMs (e.g., InternVL-2.5 (Chen et al., 2024b)), enabling fully automatic annotation for synthetic distortions and semi-automatic labeling for authentic distortions. This pipeline is scalable, reliable, and cost-effective. **(iii) Compatibility with Existing Data.** We further integrate widely-used datasets such as Q-Instruct (Wu et al., 2024a) and DQ-495K (You et al., 2025b), enabling conflict-free joint training that leads to synergistic performance improvements across tasks.

Beyond task-and-data reformulation, we focus on establishing a unified paradigm for comprehensive quality understanding, instead of pushing performance on isolated IQA tasks. This paradigm is instantiated through a simple two-stage design, which progressively extends the model from text-level multi-granularity quality reasoning to pixel-level grounding. The central challenge in achieving such a unified paradigm lies in integrating grounding while preserving text-level reasoning behavior and instruction-following capabilities. A key observation underlying our design is that many existing SAM-based LMM grounding approaches tightly couple language generation with pixel-level grounding through explicit special tokens (e.g., *<seg>*) (Chen et al., 2024a; Lai et al., 2024; Rasheed et al., 2024). While effective for specific grounding tasks, such token-level coupling introduces two fundamental limitations: (i) it alters the original language modeling space and has been shown to impair instruction-following and reasoning capabilities (Wu et al., 2024c); (ii) it requires additional task-specific fine-

tuning of the grounding module to interpret these special tokens. In contrast, we advocate a different design principle: grounding should be realized by reusing native language model outputs, rather than by introducing intrinsic modifications to the language generation process. Following this principle, we propose a text-to-point grounding paradigm that bridges textual semantics and pixel-level perception by mapping native textual logits to spatial coordinates used as point prompts. Instead of introducing special grounding tokens, the point prompts can be directly consumed by off-the-shelf segmentation models such as SAM (Kirillov et al., 2023). This paradigm is non-intrusive to the language model: it requires no architectural modification, no additional grounding supervision, and no task-specific fine-tuning of the segmentation module. As a result, it exhibits three desirable properties: *reasoning-preserving*, *training-free*, and *plug-and-play*. Together, this paradigm complements our task and data formulation, forming a unified approach to multi-granularity IQA.

In summary, our contributions are three-fold:

- **Task formulation and Multi-Granularity Dataset.** We formulate image quality assessment as a unified family of complementary tasks. Based on this formulation, we introduce IQA-Spider-33K, a dataset constructed with a scalable and reliable annotation pipeline, which emphasizes distortion diversity, rigorous task design, and compatibility with existing IQA instruction data.

- **Unified Model Design with Training-Free Grounding.** We propose a unified LMM-based framework with a two-stage design that progressively extends text-level multi-granularity quality understanding to pixel-level grounding. Moreover, we introduce a training-free text-to-point grounding paradigm, enabling grounding without the loss of reasoning capability for other tasks.

- **Benchmark Establishment and Strong Performance.** We highlight the importance of multi-granularity quality understanding and establish a new benchmark for text-level quality reasoning and pixel-level quality grounding. The strong performance over multiple benchmarks (*e.g.*, ours, Q-Bench-A1, and Q-Ground-Test) validates the effectiveness and versatility of the proposed framework.

## 2. Related Work

**LMM-based Image Quality Assessment.** The LMM-based IQA models have progressed along three dominant pathways. The first (Zhang et al., 2023b; Wang et al., 2023; Wu et al., 2023b; You et al., 2025a; Lu et al., 2025b) focuses on feature-text alignment, where LMMs are adapted to map

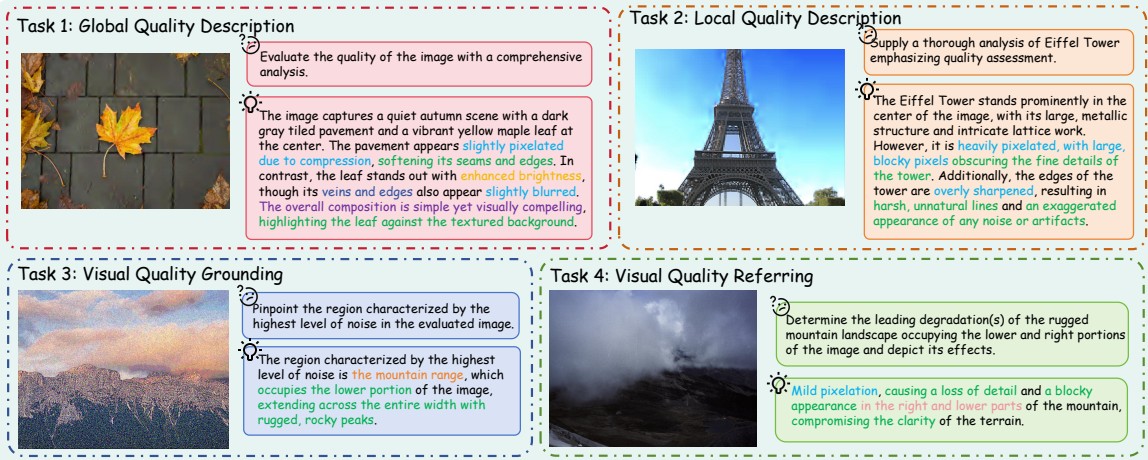

*Figure 1.* **The components of our task paradigms**, include (a) Global Quality Description, (b) Local Quality Description, (c) Visual Quality Grounding, and (d)Visual Quality Referring.

visual quality characteristics into textual representations. The second direction (Zhu et al., 2024; Wu et al., 2024d) explores specialized prompt strategies, which are designed to unlock the inherent quality perception ability of LMM. The third pathway advances explainable IQA (EIQA), especially with the integrated capability of LMMs. Early works introduce a low-level benchmark (Wu et al., 2023a) for general LMMs (Zhang et al., 2023a; Zhu et al., 2023; Chen et al., 2023b; Gao et al., 2023) and several quality-specific instruction tuning datasets (Wu et al., 2024a; You et al., 2025b; Wu et al., 2024b), focusing on both holistic quality and attribute-specific perception. Recent works (Li et al., 2025; Wen et al., 2025; Lu et al., 2025a) utilize reinforcement learning for further enhancement. To enable finer-grained understanding, Q-Ground (Chen et al., 2024a) conducts pixel-level quality grounding but compromises instruction-following and reasoning. Grounding-IQA (Chen et al., 2024c) adds quality referring via bounding boxes, but lacks general applicability. To address more realistic and diverse scenarios, we revisit these paradigms and propose a multi-task dataset supporting quality assessment from global to pixel level, without degrading chat capabilities.

**LMM-based Visual Grounding.** LMM-based grounding models can localize objects with complex reasoning during user interactions. Some methods (Bai et al., 2025b; Chen et al., 2023b; Peng et al., 2023; You et al., 2023) output bounding box coordinates through textual token generation. Text4Seg (Lan et al., 2024) casts image segmentation as text generation, which fails in dense scenarios. In contrast, several recent approaches (Lai et al., 2024; Rasheed et al., 2024; Wei et al., 2024; Xia et al., 2024) train LMMs to predict a special segmentation token (*i.e.*, $<seg>$) that represents the grounded object and guide the segmentation head (*e.g.*, SAM (Kirillov et al., 2023)) to generate masks, which can lead to degradation in instruction-following ability. Unlike

these methods, we perform grounding in a training-free manner by implicitly mapping the positional terms to point coordinates for the segmentation head. This eliminates the need for grounding instruction-tuning data while maintaining the model's original instruction-following capability.

## 3. Task Paradigms and Dataset Construction

### 3.1. Task Paradigms

To enable multi-granularity perception, we propose a four-task paradigm with a primary focus on region-level quality understanding, which is shown in Fig. 1. For better applicability, we conduct all tasks under the no-reference setting. The four tasks are introduced as follows:

- **Task 1: Global Quality Description.** Follow the existing works (You et al., 2025b; Wu et al., 2024a), we conduct this task by given a target image and output the quality-related answer. Specifically, our dataset emphasizes more detail(*e.g.*, texture damage) in this task.

- **Task 2: Local Quality Description.** This task requires the model to produce region-level quality descriptions, which calls for localized perception to first identify relevant areas, and then assess their quality factors.

- **Task 3: Visual Quality Grounding.** For this task, given an image and a question, the model is required to first produce a textual region-level answer, followed by a pixel-level segmentation mask. Furthermore, we decompose the task into three sub-tasks, each defined by a distinct grounding objective. (a) hybrid distortion intensity grounding (HyD-G). The goal of this task is to ground the region with the most prominent (*i.e.*, maximum or minimum) cumulative intensity resulting

from all types of present distortions. (b) single distortion intensity grounding (SiD-G). In contrast, this task focuses on a specific distortion within all present distortions, aiming to ground the region with the highest or lowest intensity of that particular distortion in the image. (c) distortion accumulation order grounding (DAO-G). Given a predefined distortion accumulation order, the objective of this task is to identify the region that matches both the distortion types and their accumulation order.

- **Task 4: Visual Quality Referring.** This task focuses on identifying distortion types within referred image regions. To support both concise and detailed responses, we design two answer patterns of increasing complexity. The *short-answer* pattern requires the model to accurately identify all types of distortions present in the region. The *long-answer* pattern extends this by also requiring the model to describe the perceptual effects of these distortions, enabling more comprehensive region-level understanding.

### 3.2. Dataset Construction

Following our task paradigms, we construct a unified multi-granularity dataset, IQA-Spider-33K, to support both quality reasoning and pixel-level quality grounding, which comprises two subsets: 1) The instruction-tuning subset consists of {*image*, *question*, *answer*} triplets, enabling multi-level, textual quality assessment in the first training stage. 2) The grounding subset augments these triplets with segmentation masks, supporting pixel-level supervision in the second stage. To efficiently construct this dataset at scale, we adopt a hybrid annotation pipeline that leverages multiple open-source models. We first preprocess images using SAM-based tools and human annotators to identify distortion regions. Then, we employ InternVL-2.5(Chen et al., 2024b) to generate multi-turn question & answer pairs. The detailed construction pipeline is provided in Fig. 2.

**Image Data Collection.** In this stage, we collect all necessary image data and distortion-related annotations, including region-level distorted images, distortion type and intensity, semantic region labels, segmentation masks, bounding boxes, and a brief quality description for each image. We take both authentic and synthetic distortions into consideration, which the original image data is chosen from KonIQ (Hosu et al., 2020) and KADIS-700K (Lin et al., 2019) dataset respectively. For images with authentic distortions—either uniformly distributed or localized at the region level—we employ human annotators to provide reliable ground-truth annotations. Under the human decision, the image with region-level distortions is annotated with all the image data information, and the globally distorted image is solely required to a brief description. To ensure scalability and transparency in data construction, we deliberately

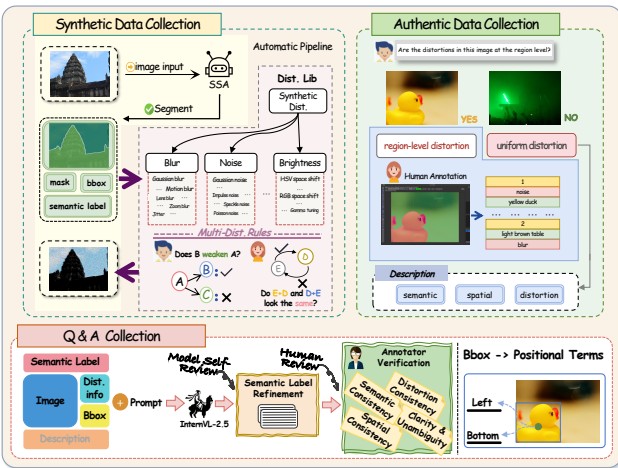

*Figure 2.* **Dataset construction pipeline.** For synthetically distorted data, we design a fully automatic pipeline that begins with SSA segmentation and proceeds with region-level hybrid distortion accumulation. For authentically distorted data, we involve human annotators to identify and categorize region-level distortions. These data are then used as prompts or image inputs for InternVL-2.5 to generate question & answer pairs.

avoid using expensive and closed-source large multimodal models (*e.g.*, GPT-4V (OpenAI, 2023)) for authentically distorted images.

For synthetic distortions, we propose an automated, human-free pipeline that generates region-level distortion data from high-quality source images. Existing datasets are often limited in distortion diversity or only provide global-level degradations, which do not satisfy our needs for fine-grained region-level quality assessment. To overcome this, we leverage the open-source tool **Semantic Segment Anything (SSA)** (Chen et al., 2023a) to extract semantic instance regions. We then apply various synthetic distortions to these regions, guided by the SSA-generated segmentation masks. This process automatically produces distorted images, region-level distortion masks, distortion type labels, and corresponding semantic region labels—**all without human annotation**. To better simulate realistic degradation patterns, we adopt the multi-distortion composition strategy proposed in (You et al., 2025b), and manually define all perceptually recognizable distortion accumulation orders in accordance with the existing constraints of the human visual system. See more details in Appendix A.1.

**Question & Answer Collection.** We collect all questions and answers through an automated pipeline, which is composed of the following steps: **i) Defining the Question Format**: We first prompt InternVL-2.5 to define the structure of the question format. This ensures that all generated questions follow a consistent and standardized schema. **ii) Multi-round Dialogue for Answer Generation**: Once the question format is established, we initiate a multi-turn dialogue with InternVL-2.5 incorporating image information.

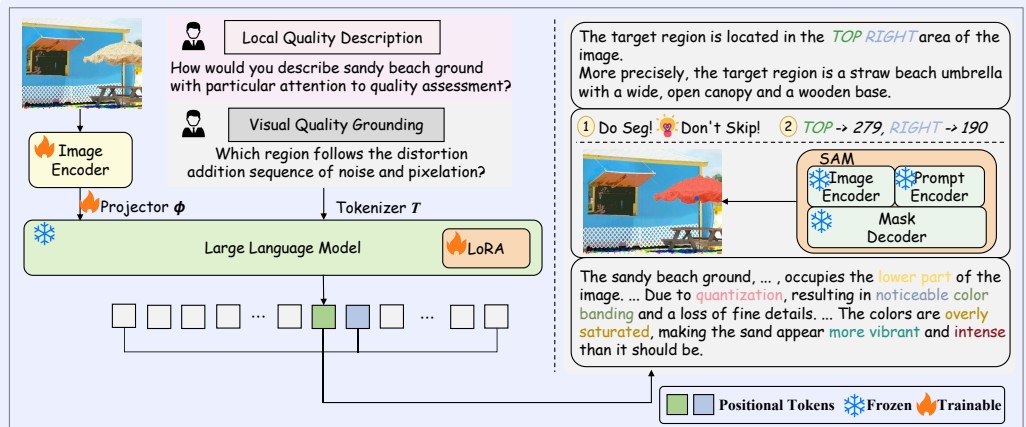

*Figure 3.* **Model framework.** An LMM with a segmentation head (*i.e.*,SAM) is employed for multi-granularity quality understanding. Text-level reasoning guides whether to perform pixel-level segmentation, triggered only under non-uniform distortions.

Through this iterative dialogue, answers are progressively generated and filled into the predefined format. **iii) Semantic Label Refinement**: To prevent incorrect semantic labels initially generated by SSA, we refine and correct the semantic labels before proceeding to answer generation. This ensures higher semantic accuracy. **iv) Grounded Answer Generation**: For grounding-related answers, we generate both spatial descriptions and region-level semantic descriptions using a set of predefined positional terms. Specifically, the terms *left* and *right* are used to indicate horizontal (x-axis) positions. And the terms *top* and *bottom* are used to indicate vertical (y-axis) positions. These positional terms are selected by converting the center of bounding box coordinates of the target object. These positional terms are mapped by the normalized coordinates of the center point of the bounding box of the target object, which is performed as the following equations:

$$T_x = \begin{cases} left, & \frac{x}{W} \in [0, \frac{1}{2}) \\ right, & \frac{x}{W} \in [\frac{1}{2}, 1] \end{cases} \qquad T_y = \begin{cases} top, & \frac{y}{H} \in [0, \frac{1}{2}) \\ bottom, & \frac{y}{H} \in [\frac{1}{2}, 1] \end{cases} \qquad (1)$$

**Dataset Verification.** To validate the reliability of our dataset, we randomly sample 40% of the data and ask 10 human annotators to rate each instance along four dimensions: semantic consistency, spatial consistency, distortion consistency, and linguistic clarity & unambiguity. Ratings are assigned on a five-point scale from 1 to 5. The results show that, for each evaluation dimension, over 80% of the sampled instances are rated as 4 or 5, indicating the high quality and reliability of our dataset. More details are provided in Appendix A.1.

## 4. Methodology

Our proposed model is designed for multi-granularity image quality perception through conversation, addressing tasks from global understanding to fine-grained region-level refer-

ring and pixel-level grounding. To achieve this, we design a unified framework that progressively enhances the model's perception capabilities across different levels. The model first learns to understand image quality at the global-level and region level through instruction tuning, then escalates to pixel-level grounding in a training-free manner via a novel text-to-point paradigm.

### 4.1. Overall Framework

Our framework consists of two main components: a LMM backbone and a segmentation head. An overview of our model is shown in Fig.3.

**LMM Backbone.** The LMM takes an image and a question as input and outputs a textual answer. The image is encoded and projected into the textual embedding space, while the question is tokenized and embedded. Both are then fed into a LLM to generate the answer. To improve efficiency and reduce redundant computation, the model determines the necessity of grounding based on the generated textual answer. If the textual answer indicates that the target region covers the entire image, grounding is skipped and the full image is used as the mask; otherwise, grounding is performed.

**Segmentation Head.** For pixel-level grounding, a training-free method (*i.e.*, text-to-point paradigm) is used to unlock LMM's grounding ability. The LMM's textual output is mapped into a point-based prompt, which is processed by a SAM-like segmentation model to generate the final mask.

### 4.2. Text-to-point paradigm

To leverage textual output to guide the grounding process, we implicitly map the positional terms to the point prompt coordinates. Specifically, with the assess of the logits from the hidden states of LMM, which reflect the probability distribution of every token, we can get the probabilities of

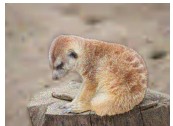

Query : Provide a careful evaluation of the meerkat with a focus on the aspects of quality.

Ours: A meerkat with a light brown coat and darker eye patches sits upright on a wooden stump in the center. The image is overly bright, washing out the meerkat's fur details. Additionally, a grainy noise effect obscures finer details of its fur and features.

DepictQA: The image depicts a meerkat sitting on a log. The evaluated image exhibits slight noise, visible as graininess throughout, which slightly reduces the clarity of the meerkat's fur and the wood texture. Additionally, there is a slight blur, softening the edges and details, particularly noticeable in the meerkat's fur and facial features. Despite these distortions, the overall composition and subject are still recognizable. The image maintains a fair level of detail, making it acceptable for contexts where high fidelity is not critical.

Q-Instruct: The image has accurate exposure with no obvious overexposed or underexposed areas. The focus is accurate, but the picture is not clear. There are noticeable noise and severe noise interference. The colors are rich, and the lighting is sufficient. The texture details are not clear. The composition is centered, with the main subject clearly being the meerkat in the middle. Therefore, the quality is poor.

Co-Instruct: The image has accurate exposure with no obvious overexposed or underexposed areas. The focus is precise, and the picture is clear with no noticeable noise. The colors are rich, and there is sufficient lighting. However, the texture details are not clear, and there is a significant amount of noise in the image. The composition is centered, and the main subject, a small animal in the center, is clearly defined. Therefore, the quality is poor.

(a) Visualization of Local Quality Description

Query : What region follows the pattern of adding pixelation before contrast strengthening in the evaluated image?

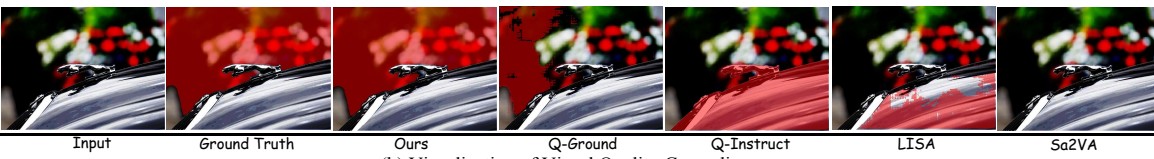

Input      Ground Truth      Ours      Q-Ground      Q-Instruct      LISA      Sa2VA

(b) Visualization of Viusal Quality Grounding

*Figure 4.* **Visualization of different methods.** (a) Visualization of Local Quality Description task. (b) Visualization of the distortion accumulation order grounding (DAO-G) sub-task in Visual Quality Grounding.

positional terms by applying a close-set softmax operation to the corresponding logits. We adopt the term *left* and *right* for horizontal position representation, and *top* and *bottom* for vertical position representation. The softmax operation is illustrated as follows:

$$p_{l_i} = \frac{e^{\chi_{l_i}/\tau}}{\sum_j e^{\chi_{l_j}/\tau}}, \quad l = \{w, h\} \qquad (2)$$

$$\{w_i|_{i=0}^1\} = \{left, right\}, \quad \{h_i|_{i=0}^1\} = \{top, bottom\} \qquad (3)$$

where $\tau$ denotes the temperature, $\chi$ is the logits. $w$ and $h$ are the collection of horizontal position terms and vertical position terms, respectively. We define the normalized x-coordinate and y-coordinate values of an image from 0 to 1. With the rule that coordinate origin is the top left corner, the terms *left* and *right* can be mapped to value 0 and 1 respectively. Similarly, *top* can be mapped to value 0 and *bottom* can be mapped to value 1. Finally, we calculate the point coordinates using a weighted average and then scale them back to the original size. The process is shown as:

$$x = \sum i p_{w_i} \times W, \quad y = \sum i p_{h_i} \times H \qquad (4)$$

where $W$ and $H$ represent the width and height of the original image.

Once we get the point coordinates, we adopt it as prompt for segmentation without more supervision. Unlike the previous works, that generate the prompt explicitly by training the LMM to output the special token, we implicitly convert text into point prompt, keeping the textual output format unaltered. Moreover, compared to other methods (Wu et al., 2024c; Cao et al., 2024) that implicitly generate mask prompt by attention map, which either cause high memory overhead or require an additional image encoder. We bridge the gap of LMM and segmentation model in a simple way and can be adopt widely to any LMM.

### 4.3. Hybrid Dataset Training

To stimulate the model with strong and generalizable quality perception ability, we scale up the training dataset by taking two existing dataset partially with our own dataset. Our training dataset consists of three parts. i) The Q-Instruct(Wu et al., 2024a) dataset, which contains global and local quality question answering, includes the low-level visual perception tasks. ii) The DQ-495K(You et al., 2025b) dataset. A collection of distortion identification tasks and assessment reasoning tasks. It is helpful for enhancing the distortion perception and visual damage description capabilities. iii) Ours dataset, which contains four tasks, inspiring the fine-grained perception ability of the models. We believe that the combination of these datasets contributes to the enhancement of quality-centric understanding abilities and conversational ability, enabling it to unify a wide range of tasks.

**Training Objectives.** The segmentation head is weight-frozen without specific fine-tuning. As for base LMM, we apply the instruction tuning to make the base LMM be quality-aware. Following most supervised fine-tuning works (Yin et al., 2023; Liu et al., 2023b) in LMMs, we apply the next token prediction loss as our training objective, which is a cross-entropy loss for token generation.

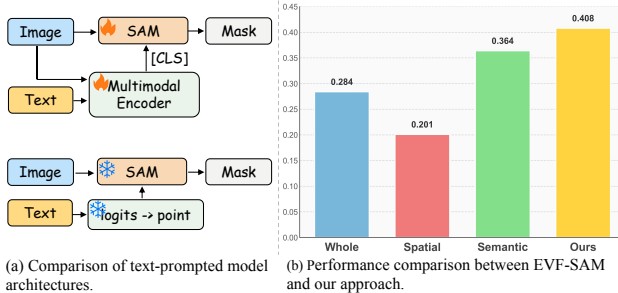

(a) Comparison of text-prompted model architectures.

(b) Performance comparison between EVF-SAM and our approach.

*Figure 5.* **Comparison between EVF-SAM and our approach.** (a) Comparison of two text-prompted model architectures. (b) Grounding performance comparison on our dataset.

## 5. Experiment

### 5.1. Experimental Setup

**Implementation Details.** We use three base models: Phi-3.5-Vision (4B) (Abdin et al., 2024), Qwen2.5-VL (7B) (Bai et al., 2025b) and Qwen3-VL (7B) (Bai et al., 2025a). Following previous works(Lai et al., 2024; Rasheed et al., 2024), we adopt SAM as the segmentation head, enabling the grounding ability of the model. In the fine-tuning stage, we apply the parameter-efficient technique LoRA (Hu et al., 2022) for LLM, and apply full-tuning for the visual encoder as well as the visual projector.

### 5.2. Results

**Text-level Reasoning Tasks: Quantitative Results of Our Benchmark.** As shown in Tab. 1, we evaluate both quality-specific and open-source models on our benchmark. Our model achieves the best performance across all tasks, demonstrating its strong capability for multi-granularity quality understanding. Other models exhibit small performance gap on the global description task, suggesting a certain level of global quality perception. However, the gap becomes larger in fine-grained tasks, indicating their limited ability to capture fine-grained quality details. Further analysis in Fig. 4(a) reveals that existing IQA-specific models: (i) fail to respond to the local targets specified in the query or to respect the intended granularity, often degenerating into global-level descriptions of the entire image, and (ii) fail to accurately recognize local distortions. These results further indicate the limited fine-grained quality perception of existing models.

**Text-level Reasoning Tasks: Quantitative Results of Q-Bench.** We evaluate the general low-level reasoning capability of our model on Q-Bench-A1 in a multiple-choice question answering setting without any task-specific training. As shown in Tab. 2, our model outperforms all baselines, indicating comprehensive low-level quality understanding.

**Pixel-level Grounding Task: Quantitative Results of Our Benchmark.** In Tab. 3, we evaluate the pixel-level quality

grounding task with two general LMM-based grounding methods and two quality-aware models, namely Q-Ground and Q-Instruct. Notably, we adapt Q-Instruct with the text-to-point paradigm to activate its grounding capability. Traditional non-LMM segmentation approaches are excluded from our evaluation, as they lack the capacity for the complex and implicit reasoning required by our benchmark. The results demonstrate that general LMM-based grounding models exhibit limited performance due to their insufficient quality-aware visual perception under severe distortions and the absence of quality-specific knowledge in the language model. In contrast, our model achieves the best performance, with Q-Instruct ranking second, outperforming both general-purpose and quality-specific grounding methods. Furthermore, the visualization in Fig.4 (b) demonstrates that other models generate incorrect or empty masks, underscoring the superior pixel-level quality grounding capability of our method.

*Table 3.* **Visual quality grounding evaluation on our benchmark.** The results of Q-Ground are obtained from our own reproduction.

| Model/Dataset | DAO-G | HyD-G | SiD-G | Average |
|---|---|---|---|---|
| LISA(7B) (Lai et al., 2024) | 0.063 | 0.101 | 0.086 | 0.078 |
| Sa2VA(8B) (Yuan et al., 2025) | 0.186 | 0.194 | 0.209 | 0.192 |
| Q-Instruct (Wu et al., 2024a) | 0.243 | 0.315 | 0.229 | 0.264 |
| Q-Ground* (Chen et al., 2024a) | 0.192 | 0.176 | 0.202 | 0.190 |
| Ours (Phi-3.5-Vision) | 0.375 | 0.354 | **0.342** | 0.364 |
| Ours (Qwen2.5-VL) | 0.402 | 0.363 | 0.341 | 0.381 |
| Ours (Qwen3-VL) | **0.439** | **0.396** | 0.313 | **0.408** |

**Pixel-based Grounding Task: Quantitative Results of Q-Ground-Test.** In Tab. 5, we evaluate our model on Q-Ground-Test. We adopt the same prompt format as in training to generate positional terms for point mapping. Unlike Q-Ground, our model does not rely on quality-related textual references in the prompt. Notably, our grounding stage is training-free, and no Q-Ground-100K data is used in the first stage. Nevertheless, our model still outperforms all baselines that are fine-tuned on Q-Ground-100K. This further demonstrates the effectiveness and strong generalization ability of our model.

**Visual Quality Scoring Task.** In Tab. 6, we assess the visual scoring ability by testing on the synthetic datasets KADID-10K (Lin et al., 2019). Despite the absence of task-specific training, our model achieves comparable performance to other models within the description-oriented category (*e.g.*, Q-Instruct).

### 5.3. Ablative Study

**Effectiveness of Hybrid Datasets Training.** As shown in Tab. 4, we conduct an ablation study to analyze the impact of hybrid dataset training by progressively incorporating different data sources. For description tasks, incorporating any single additional dataset leads to a slight performance drop, while combining all datasets yields the best performance.

*Table 1.* **Quantitative comparison on our benchmark.** 5 round GPT-4V score is adopted as metric for description and grounding tasks, with a scale of 0-10 and 0-5 respectively. Accuracy is used to evaluate quality referring task.

| Model | Global Des. | Local Des. | Grounding | Ref$^{short}$ | Ref$^{long}$ |
|---|---|---|---|---|---|
| Q-Instruct (Wu et al., 2024a) | 3.80 | 3.88 | 0.93 | 0.116 | 0.129 |
| Co-Instruct (Wu et al., 2024b) | 3.50 | 3.53 | 0.52 | 0.121 | 0.119 |
| DepictQA-Wild (You et al., 2024) | 3.49 | 4.16 | 0.24 | 0.196 | 0.164 |
| Phi-3.5-Vision (Abdin et al., 2024) | 4.03 | 3.01 | 0.68 | 0.100 | 0.101 |
| Qwen2.5-VL(7B) (Bai et al., 2025b) | 3.30 | 3.21 | 1.25 | 0.087 | 0.117 |
| Qwen3-VL(7B) (Bai et al., 2025a) | 5.90 | 5.45 | 0.70 | 0.126 | 0.176 |
| Ours (Phi-3.5-Vision) | 6.17 | 6.55 | 2.29 | 0.321 | 0.290 |
| Ours (Qwen2.5-VL) | 6.75 | 6.88 | 2.13 | 0.513 | 0.416 |
| Ours (Qwen3-VL) | **7.12** | **7.10** | **2.41** | **0.594** | **0.484** |

*Table 2.* **Quantitative comparison on LLVisionQA-dev dataset from Q-Bench-A1.**

| Model | Accuracy |
|---|---|
| random guess | 37.80% |
| LLaVA-v1.5(7B) (Liu et al., 2023a) | 58.66% |
| LLaVA-v1.5(13B) (Liu et al., 2023a) | 62.14% |
| InternLM-XComposer-VL (Zhang et al., 2023a) | 65.35% |
| mPLUG-Owl2 (Ye et al., 2024) | 61.61% |
| Q-Instruct (Wu et al., 2024a) | 67.56% |
| Ours (Phi-3.5-Vision) | 70.30% |
| Ours (Qwen2.5-VL) | 73.24% |
| Ours (Qwen3-VL) | **74.45%** |

*Table 4.* **Ablation study on hybrid datasets based on Qwen3-VL.**

| | I | II | III | IV |
|---|---|---|---|---|
| Ours | ✓ | ✓ | ✓ | ✓ |
| Q-Instruct | | ✓ | | ✓ |
| DQ-495K | | | ✓ | ✓ |
| Global Des. | 7.01 | 6.99 | 7.00 | **7.12** |
| Local Des. | 7.07 | 7.03 | 6.86 | **7.10** |
| Grounding | 2.42 | **2.53** | 2.36 | 2.41 |
| Ref$^{short}$ | 0.541 | 0.542 | 0.547 | **0.594** |
| Ref$^{long}$ | 0.458 | 0.466 | 0.476 | **0.484** |

*Table 5.* **Visual quality grounding evaluation measured by mIoU.**

| Model/Dataset | Q-Ground-Test |
|---|---|
| LISA (Lai et al., 2024) | 0.227 |
| PixelLM (Ren et al., 2024) | 0.252 |
| Q-ground (Chen et al., 2024a) | 0.271 |
| Ours (Phi-3.5-Vision) | 0.293 |
| Ours (Qwen2.5-VL) | 0.295 |
| Ours (Qwen3-VL) | **0.338** |

*Table 6.* **Quantitative comparison over artificial visual scoring datasets (SRCC/PLCC).**

| Model / Dataset | KADID-10K |
|---|---|
| MUSIQ (Ke et al., 2021) | 0.556/0.575 |
| CLIP-IQA+ (Wang et al., 2023) | 0.654/0.653 |
| ManIQA (Yang et al., 2022) | 0.465/0.499 |
| Q-Instruct (Wu et al., 2024a) | 0.698/0.676 |
| Ours (Phi-3.5-Vision) | **0.815/0.783** |
| Ours (Qwen2.5-VL) | 0.706/0.711 |
| Ours (Qwen3-VL) | 0.741/0.746 |

For grounding, although the performance slightly fluctuates across different settings, the model achieves the best results when trained with our dataset and Q-Instruct. This suggests that grounding primarily benefits from datasets with explicit alignment between visual target regions and textual descriptions, while general low-level knowledge from Q-Instruct further provides complementary improvements. Additionally, the referring tasks benefit the most from hybrid training. Both referring$^{short}$ and referring$^{long}$ show consistent gains as more diverse data are introduced, achieving the best results in setting IV. Overall, these results demonstrate that combining complementary datasets effectively enhances the model's ability for multi-granularity quality understanding.

**Effectiveness of Text-to-point paradigm.** As for text-prompted SAM architectures, we compare our text-to-point paradigm with EVF-SAM (Zhang et al., 2024), which links the text and the segmentation model(*i.e.*, SAM) by adopting a trainable multimodal encoder to generate prompt. The comparison of the model architectures is illustrated in Fig. 5 (a). The text-level grounding output consists of two parts: a spatial answer and a semantic answer. Based on this structure, we evaluate EVF-SAM using three types of textual inputs: the *whole* answer, the *spatial* component only, and the *semantic* component only. As shown in Fig. 5 (b), EVF-SAM achieves its best performance when prompted with the semantic text alone, and its performance degrades when using the whole answer. In contrast to EVF-SAM, which is jointly fine-tuned with a multimodal encoder and SAM and leverages rich semantic prompts, our method is training-free and relies on only minimal information, yet achieves better performance. These results validate both

the effectiveness and efficiency of our proposed grounding paradigm.

# 6. Conclusion

In this study, we introduce a novel four-task paradigm emphasizing region-level quality understanding. To support it, we construct a unified multi-task dataset, IQA-Spider-33K, encompassing global and local description, pixel-level grounding and region-level referring, enabling multi-granularity perception. Furthermore, we present the IQA-Spider framework, which adopts a two-stage optimization strategy: it first optimizes textual perception tasks, then transitions to pixel-level perception through the text-to-point grounding paradigm. Our work establishes a new benchmark for fine-grained quality assessment tasks and holds promise for broader applications in quality-related domains.

# Acknowledgements

This work was supported in part by NSFC under Grant 62371434, U25B2010 and 623B2098, the Fundamental Research Funds for the Central Universities (No. WK2100250064), Anhui Postdoctoral Scientific Research Program Foundation (No.2025A1015), the Postdoctoral Fellowship Program of CPSF under Grant Number GZC20252293, the China Postdoctoral Science Foundation-Anhui Joint Support Program under Grant Number 2024T017AH, and China Postdoctoral Science Foundation under Grant Number 2025M783529. We also thank our colleagues for helpful discussions and valuable feedback.

## Impact Statement

This paper presents work whose goal is to advance the field of Machine Learning. There are many potential societal consequences of our work, none of which we feel must be specifically highlighted here.

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

# A. Appendix

## A.1. Dataset Details

**Multi-distortion accumulation setting.** In scenarios involving multiple distortions, the accumulation order becomes unidentifiable when different orders produce identical visual outcomes. To make up for this, we manually define the distinguishable distortion accumulation orders. Following DQ-495K (You et al., 2025b), we first determine the distortion combinations by removing those exhibiting similar or contradictory effects, ensuring the distortion categories clear and distinguishable. Secondly, we enumerate all possible orders and filter the recognizable ones, which is shown in Tab. 8.

*Table 7.* **Statistics of four tasks in our dataset.**

| Tasks | Global Des. | Local Des. | Grounding | Ref$^{short}$ | Ref$^{long}$ |
|---|---|---|---|---|---|
| Train | 3251 | 9742 | 11669 | 4506 | 4222 |
| Test | 629 | 460 | 586 | 780 | 786 |

**Details of Dataset Construction.** We construct synthetically distorted images by using the pristine images from Kadis-700K (Lin et al., 2019), and we leverage human to annotate the authentically distorted images from KonIQ (Hosu et al., 2020). The image source statistics are illustrated in Fig. 6. For synthetic distortions, we use the distortions categories from DQ-495K (You et al., 2025b). For human annotations, the visualized wordcloud of distortion categories is shown in Fig. 7. Generally, the statistics of four tasks in our dataset in demonstrated in Tab. 7. All the question templates are shown in Tab. 10, Tab. 11, Tab. 12, Tab. 13, Tab. 14, Tab. 15, Tab. 16, Tab. 17 and Tab. 18.

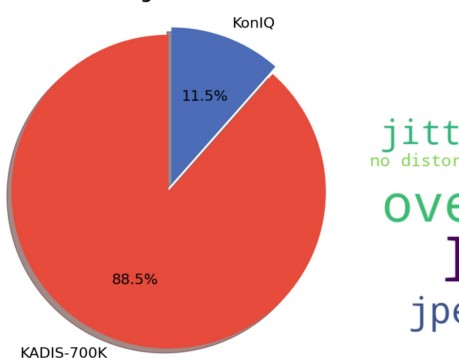

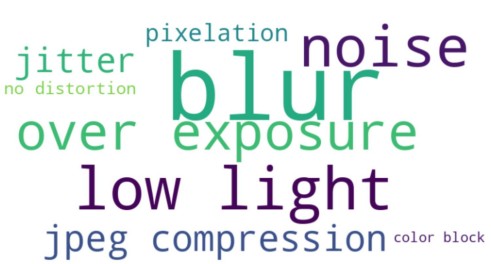

*Figure 6.* **Image source statistics.**

*Figure 7.* **Wordcloud map of human annotated distortion categories.**

**Details of Dataset Verification.** We first define a five-point rating scale for the verification process, where scores range from 1 to 5. A score of 1 indicates the lowest quality, while a score of 5 represents the highest quality. To promote inter-annotator consistency, we provide the 10 annotators with standardized examples for each rating level as a calibration reference. After the verification process, we aggregate the ratings for each evaluation dimension, as shown in Fig. 8, Fig. 9, Fig. 10, and Fig. 11.

*Table 8.* **Distinguishable distortion accumulation orders.**

| First Distortion | All Possible Second Distortions |
|---|---|
| Blur | Compression, Noise |
| Compression | Blur, Noise |
| Contrast Weaken | Noise, Blur, Compression, Contrast Weaken, Pixelate, Saturate Weaken |
| Pixelate | Noise |
| Saturate Weaken | Noise |

## A.2. Reliability Evaluation of GPT Score

To validate the reliability of GPT Score, we conduct a preliminary study to examine its alignment with human evaluations on global description, local description, and grounding task. We randomly sample 100 instances for each task. Each generated

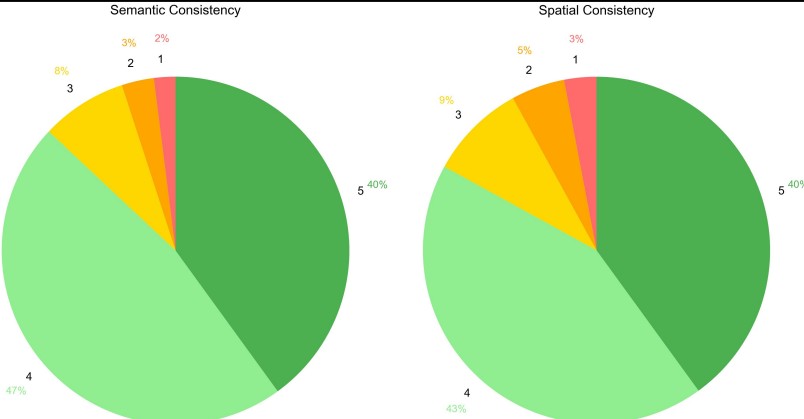

*Figure 8.* **Semantic consistency verification results.**

*Figure 9.* **Spatial consistency verification results.**

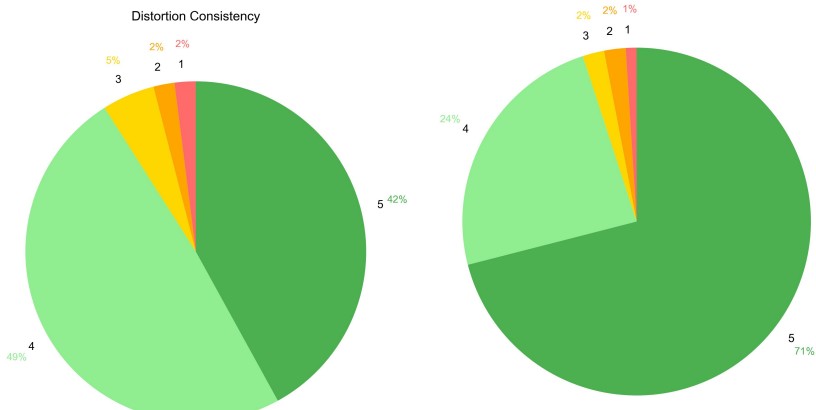

*Figure 10.* **Distortion consistency verification results.**

*Figure 11.* **Linguistic clarity & unambiguity consistency verification results.**

answer is rated by 5 human annotators first in order to measure inter-annotator agreement using the intra-class correlation coefficient (ICC). We then compute GPT Score for the same set of instances using a fixed prompt with temperature set to zero. The alignment between GPT Score and human ratings is evaluated using SRCC and PLCC metrics. The results in Tab. 9 show a strong and consistent correlation between GPT Score and human judgments, supporting its use in our final evaluation.

## A.3. More Results

As the visualized results of local quality description and visual quality grounding are illustrated in the main manuscript, we demonstrated more qualitative results of global quality description and visual quality referring in Fig. 12 and Fig. 13. In global quality description task, our model output is more detailed, which can describe the distortion type and visual effect with fine-grained spatial location. For visual quality referring task, our model not only identifies distortions at the region level but also explains their visual impact, showing its strong region-level quality understanding capability. These results further validate the comprehensive multi-granularity perception capability of our model.

*Table 9.* **Alignment between GPT Score and human ratings.**

| Task | ICC ↑ | SRCC ↑ | PLCC ↑ |
|---|---|---|---|
| Global description | 0.78 | 0.76 | 0.75 |
| Local description | 0.75 | 0.69 | 0.66 |
| Grounding | 0.74 | 0.68 | 0.67 |

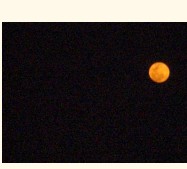

The image captures a night scene dominated by a dark, starless sky, with a bright, orange-yellow moon positioned towards the upper right corner. The sky, which forms the background, is heavily marred by noise, resulting in a grainy and speckled texture that disrupts the smoothness of the dark backdrop. This noise effect makes the sky appear less uniform and more chaotic. The moon, while bright and prominent, is also affected by noise, giving it a grainy texture that slightly obscures its surface details. Additionally, the moon is blurred, softening its edges and making it appear less sharp and defined. The overall impact of these distortions is a loss of clarity and detail, with both the sky and the moon suffering from a lack of sharpness and smooth texture. The image's quality is significantly compromised by the heavy noise and blurring effects, resulting in a visually disrupted and less coherent scene.

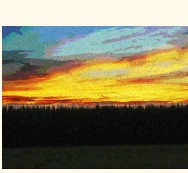

The image captures a serene sunset scene with a sky filled with warm hues of orange, yellow, and pink, transitioning into a darker blue. The sky, occupying the upper portion, is marred by a quantization effect, causing the colors to appear in distinct bands and giving it a posterized look. Additionally, the presence of noise introduces random speckles and graininess, disrupting the smooth gradient and adding a textured, gritty appearance. Below the sky, a dense line of dark green trees forms a horizon, but their details are lost due to a weakened contrast, making them appear flat and lacking depth. The lower portion of the image features a field of green grass, which is also affected by the weakened contrast, resulting in a washed-out and less vibrant appearance. The overall image structure is disrupted by the various distortions, leading to a loss of detail and a compromised visual experience. The sky's vibrant colors are dulled by the quantization and noise, while the trees and grass suffer from reduced contrast, making the scene appear less lively and dynamic.

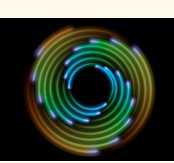

The image features a central abstract art piece with a concentric circular pattern, displaying a gradient of colors ranging from green to blue and yellow. The colors are vibrant but significantly blurred, giving the artwork a soft and diffused appearance. This central element is surrounded by a dark, almost black background that occupies the entire periphery of the image. The background is uniformly dark with no distinct features, providing a stark contrast to the colorful artwork. The low light level in the background enhances its obscurity, making it difficult to discern any details. The overall image structure is simple, with the abstract art drawing the viewer's attention due to its bright colors and central placement, while the dark background serves to isolate and highlight the artwork. The distortions, particularly the blur and low light, significantly impact the image's clarity and detail, resulting in a somewhat abstract and dreamlike quality.

*Figure 12.* **Results of global quality description task.**

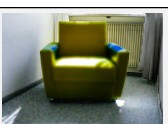

Can you list two primary distortions of the light gray window curtain positioned on the right side of the image? Answer the question and further characterize distortions effects.

Moderate pixelation, making the curtain's texture appear blocky and less defined. Significant darkening, reducing the light color to near black and diminishing its translucency.

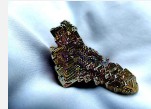

Determine the leading degradation(s) of the multicolored crystalline structure positioned in the center of the image and further depict the effects.

Moderate oversharpening, creating a harsh and unnatural appearance with exaggerated crystal edges and enhanced contrast, making the structure appear more defined and textured.

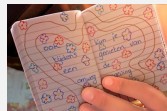

What are the two leading distortions of the hand holding the notebook in the bottom right corner? Answer the question using short phrases.

Mild blur, moderate brightening.

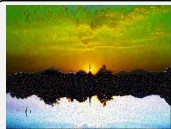

What are the two most severe degradations of the cloudy sky occupying the upper portion of the image? Answer the question using short phrases.

Moderate saturation strengthening, severe quantization.

*Figure 13.* **Results of Visual Quality Referring task.**

*Table 10.* **Question pool for local quality description task.**

| # | Question |
|---|---|
| 1 | Conduct a detailed description about {} with particular attention to quality evaluation. |
| 2 | Deliver a thorough description of {} with an emphasis on quality assessment. |
| 3 | Offer a detailed analysis of {} with a focus on evaluating its quality. |
| 4 | Supply a thorough analysis of {} emphasizing quality assessment. |
| 5 | Conduct a detailed analysis of {} that prioritizes quality evaluation. |
| 6 | Create a complete evaluation of {} with particular attention to quality. |
| 7 | Conduct a thorough evaluation of {} that prioritizes quality insights. |
| 8 | Provide a careful evaluation of {} with attention to quality assessment. |
| 9 | Create a meticulous review of {} with a focus on its quality aspects. |
| 10 | Write a thorough assessment of {} that underscores quality evaluation. |
| 11 | Conduct a thorough analysis of {} that highlights quality assessment. |
| 12 | Present an extensive evaluation of {} with a focus on the aspects of quality. |
| 13 | Conduct an in-depth appraisal of {} that considers quality-related factors. |
| 14 | Furnish a detailed evaluation of {} with insights on quality-related factors. |
| 15 | Can you offer an in-depth description of {} that highlights quality evaluation? |
| 16 | How would you characterize {} while focusing on quality evaluation? |
| 18 | Could you provide a description of {} that highlights its quality aspects? |

*Table 11.* **Question pool for global quality description task.**

| # | Question |
|---|----------|
| 1 | Assess the quality of the image with a detailed explanation. |
| 2 | Analyze the image quality and provide a thorough explanation. |
| 3 | Examine the quality of the image with an in-depth discussion. |
| 4 | Evaluate the quality of the image with a comprehensive analysis. |
| 5 | Provide an evaluation of the image quality with a complete explanation. |
| 6 | Appraise the quality of the image with a comprehensive overview. |
| 7 | Deliver a thorough evaluation of the quality of the image, highlighting both strengths and weaknesses. |
| 8 | Present an in-depth analysis of the quality of the image, addressing its advantages and areas needing improvement. |
| 9 | Offer a detailed assessment of the image quality, encompassing both positive aspects and opportunities for enhancement. |
| 10 | Conduct a comprehensive review of the quality of the image, noting strengths as well as potential improvements. |
| 11 | Present a detailed analysis of the image quality, focusing on its strong points and areas that could be enhanced. |
| 12 | Present a complete assessment of the image quality, addressing its advantages and identifying areas for improvement. |
| 13 | Offer a thorough analysis of the image quality, mentioning both favorable aspects and areas to be enhanced. |
| 14 | Present a holistic assessment of the quality of the image, detailing its strengths and areas that require enhancement. |
| 15 | Conduct a detailed review of the quality of the image, focusing on both its strong features and areas that could benefit from refinement |
| 16 | Deliver a comprehensive analysis of the quality of the image, concentrating on its merits and areas needing improvement |
| 17 | Investigate the quality of the image while considering aspects that contribute to its degradation. |
| 18 | Consider the factors affecting clarity as you assess the quality of the image. |
| 19 | Analyze the quality of the image while examining factors that lead to its degradation. |
| 20 | Investigate the image quality while evaluating the factors that result in its degradation. |
| 21 | How do you assess the quality of the image, and what aspects contribute to your opinion? |
| 22 | What are your thoughts on the quality of the image? Please elaborate on your perspective. |
| 23 | How would you evaluate the quality of the image? Share a detailed explanation of your opinion. |
| 24 | What is your perspective on the quality of the image? Expand on your evaluation. |
| 25 | Can you deliver an in-depth evaluation of the quality of the image? |
| 26 | Could you conduct a complete evaluation of the quality of the image? |

*Table 12.* **Question pool for hybrid distortion intensity grounding task.**

| # | Question |
|---|----------|
| 1 | which is the most degraded region in the evaluated image? |
| 2 | Which region exhibits the highest impact from distortions among all the regions in the evaluated image? |
| 3 | Which region shows the most severe degradation across all the regions in the evaluated image? |
| 4 | Which region suffers the most degradation compared to the other regions in the evaluated image? |
| 5 | Among all the regions in the evaluated image, which is the most significantly degraded one? |
| 6 | Which region has the lowest quality among all the regions due to distortions in the evaluated image? |
| 7 | Which region exhibits the lowest quality as a result of distortions in the evaluated image? |
| 8 | What region shows the greatest decline in quality because of distortions in the evaluated image? |
| 9 | Which region is marked by the poorest quality due to distortions in the examined image? |
| 10 | Pinpoint the region with the worst distortion in the evaluated image. |
| 11 | Determine the region that is most degraded in the evaluated image. |
| 12 | Identify the region with the highest level of distortion in the evaluated image. |
| 13 | which is the least degraded region in the evaluated image? |
| 14 | Which region is least affected by distortions compared to others regions in the evaluated image? |
| 15 | Which region has experienced the minimal level of degradation in the evaluated image? |
| 16 | Which region exhibits the lowest degree of degradation compared to all the other regions in the evaluated image? |
| 17 | Among all regions in the evaluated image, which one has the best quality with minimal distortion influence? |
| 18 | What is the region with the highest quality and minimal distortion effects in the evaluated image? |
| 19 | Which region has the highest quality with minimal impact from distortion in the evaluated image? |
| 20 | In terms of quality, which region is the least affected by distortion in the evaluated image? |
| 21 | Which region demonstrates the best quality in the evaluated image? |
| 22 | Find the region that shows the least amount of degradation in the evaluated image. |
| 23 | Determine the region that is least degraded in the evaluated image. |
| 24 | Identify the region with the lowest level of distortion in the evaluated image. |

*Table 13.* **Question pool for single distortion intensity grounding task.**

| # | Question |
|---|----------|
| 1 | Which region shows the highest level of {} in the evaluated image? |
| 2 | Which region shows the most severe {} in the evaluated image? |
| 3 | What region has the highest amount of {} in the evaluated image? |
| 4 | Which region experiences the highest degree of {} in the evaluated image? |
| 5 | Which region has the greatest level of {} in the evaluated image? |
| 6 | Pinpoint the region characterized by the highest level of {} in the evaluated image. |
| 7 | Determine the region with the highest intensity of {} in the evaluated image. |
| 8 | Which region has the most substantial {} effect in the evaluated image. |
| 9 | What region exhibits the least amount of {} in the evaluated image? |
| 10 | What is the region with the minimal level of {} in the evaluated image? |
| 11 | Which region exhibits the lowest degree of {} in the evaluated image? |
| 12 | Which region ranks the lowest in terms of {} in the evaluated image? |
| 13 | Which region demonstrates the least extent of {} in the evaluated image? |
| 14 | Identify the region that has the minimal {} in the evaluated image. |
| 15 | Which region has the most negligible {} effect? |
| 16 | Determine the region with the lowest intensity of {} in the evaluated image. |

*Table 14.* **Question pool for distortion accumulation order grounding task.**

| # | Question |
|---|----------|
| 1 | Which region follows the distortion addition sequence of {} and {} in the evaluated image? |
| 2 | Which region add {} first, followed by {} in distortion addition process in the evaluated image? |
| 3 | Identify the region that follows the {}-first, {}-second distortion addition pattern in the evaluated image. |
| 4 | What region follows the pattern of adding {} before {} in the evaluated image? |
| 5 | Determine the region that corresponds to the distortion addition order of {}, then {} in the evaluated image. |
| 6 | Which region matches the pattern of distortion addition that begins with {} and ends with {} in the evaluated image? |
| 7 | Which region begins the distortion addition process with {} in the evaluated image? |
| 8 | Which region adds {} at the beginning of the distortion sequence in the evaluated image? |
| 9 | Which region integrates {} first during the distortion addition process in the evaluated image? |
| 10 | Identify the region that brings in {} first in the distortion addition process in the evaluated image. |
| 11 | What region initiates the distortion sequence with the addition of {} in the evaluated image? |
| 12 | Determine the region that includes {} first in the distortion addition process within the evaluated image. |
| 13 | Which region integrates {} last during the distortion addition process in the evaluated image? |
| 14 | Which region incorporates {} as the last element in the distortion addition process in the evaluated image? |
| 15 | What region of the evaluated image adds {} as the final step in the distortion sequence? |
| 16 | Which region adds {} at the end of the distortion sequence in the evaluated image? |
| 17 | What region finishes the distortion sequence by adding {} in the evaluated image? |
| 18 | Determine the region that includes {} last in the distortion addition process within the evaluated image. |

*Table 15.* **Question pool for visual quality referring task** about single distortion in the short answer setting.

| # | Question |
|---|---|
| 1 | Identify the most critical one distortion of {}. Answer the question using short phrases. |
| 2 | Pinpoint the foremost image quality issue in the evaluated image. Answer the question using short phrases. |
| 3 | List the most significant distortion related to {}. Answer the question using short phrases. |
| 4 | Can you list one primary distortion of {}? Answer the question using short phrases. |
| 5 | What is the leading distortion of {}? Answer the question using short phrases. |
| 6 | In terms of image quality, what is the most glaring issue of {}? Answer the question using short phrases. |
| 7 | What is the most severe degradation of {}? Answer the question using short phrases. |
| 8 | Pinpoint the foremost image quality issue(s) of {}. Answer the question using short phrases. |
| 9 | What distortion(s) most detrimentally affect the overall quality of {}? Answer the question using short phrases. |
| 10 | What distortion(s) are most prominent when examining {}? Answer the question using short phrases. |
| 11 | What distortion(s) are most apparent of {}? Answer the question using short phrases. |
| 12 | What distortion(s) stand out of {}? Answer the question using short phrases. |
| 13 | Identify the most critical distortion(s) of {}. Answer the question using short phrases. |
| 14 | Determine the leading degradation(s) of {}. Answer the question using short phrases. |

*Table 16.* **Question pool for visual quality referring task** about single distortion in the long answer setting.

| # | Question |
|---|---|
| 1 | Identify the most critical distortion of {} and depict its effects. |
| 2 | Pinpoint the foremost image quality issue in the evaluated image and elaborate on its effects. |
| 3 | List the most significant distortion related to {} and describe its effects. |
| 4 | Can you list one primary distortion of {} and detail its effects? |
| 5 | What is the leading distortion of {}? Answer the question and describe its effects. |
| 6 | In terms of image quality, what is the most glaring issue of {}? Answer the question and elaborate on its effects. |
| 7 | What is the most severe degradation of {}? Answer the question and characterize its effects. |
| 8 | Pinpoint the foremost image quality issue(s) of {} and describe its effects. |
| 9 | What distortion(s) most detrimentally affect the overall quality of {}? Answer the question and elaborate on its effects. |
| 10 | What distortion(s) are most prominent when examining {}? Answer the question and depict its effects. |
| 11 | What distortion(s) are most apparent of {}? Answer the question and detail its effects. |
| 12 | What distortion(s) stand out of {}? Answer the question and describe its effects. |
| 13 | Identify the most critical distortion(s) of {} and elaborate on its effects.Identify the most critical distortion(s) of {} and elaborate on its effects. |
| 14 | Determine the leading degradation(s) of {} and depict its effects. |

*Table 17.* **Question pool for visual quality referring task** about multi-distortions in the short answer setting.

| # | Question |
|---|---|
| 1 | Identify two most critical distortions of {}. Answer the question using short phrases. |
| 2 | Pinpoint two foremost image quality issues in the evaluated image. Answer the question using short phrases. |
| 3 | List two most significant distortions related to {}. Answer the question using short phrases. |
| 4 | Can you list two primary distortions of {}? Answer the question using short phrases. |
| 5 | What are the two leading distortions of {}? Answer the question using short phrases. |
| 6 | In terms of image quality, what are the two most glaring issues of {}? Answer the question using short phrases. |
| 7 | What are the two most severe degradations of {}? Answer the question using short phrases. |
| 8 | Pinpoint the foremost image quality issue(s) of {}. Answer the question using short phrases. |
| 9 | What distortion(s) most detrimentally affect the overall quality of {}? Answer the question using short phrases. |
| 10 | What distortion(s) are most prominent when examining {}? Answer the question using short phrases. |
| 11 | What distortion(s) are most apparent of {}? Answer the question using short phrases. |
| 12 | What distortion(s) stand out of {}? Answer the question using short phrases. |
| 13 | Identify the most critical distortion(s) of {}. Answer the question using short phrases. |
| 14 | Determine the leading degradation(s) of {}. Answer the question using short phrases. |

*Table 18.* **Question pool for visual quality referring task** about multi-distortions in the long answer setting.

| # | Question |
|---|---|
| 1 | Identify two most critical distortions of {} and describe their effects. |
| 2 | Pinpoint two foremost image quality issues in the evaluated image and describe their effects. |
| 3 | List two most significant distortions related to {} and describe their effects. |
| 4 | Can you list two primary distortions of {}? Answer the question and characterize their effects. |
| 5 | What are the two leading distortions of {}? Answer the question and explain their effects. |
| 6 | In terms of image quality, what are the two most glaring issues of {}? Answer the question and elaborate on their effects. |
| 7 | What are the two most severe degradations of {}? Answer the question and elaborate on their effects. |
| 8 | Pinpoint the foremost image quality issue(s) of {} and depict their effects. |
| 9 | What distortion(s) most detrimentally affect the overall quality of {}? Answer the question and detail their effects. |
| 10 | What distortion(s) are most prominent when examining {}? Answer the question and describe their effects. |
| 11 | What distortion(s) are most apparent of {}? Answer the question and depict their effects. |
| 12 | What distortion(s) stand out of {}? Answer the question and describe their effects. |
| 13 | Identify the most critical distortion(s) of {} and detail their effects. |
| 14 | Determine the leading degradation(s) of {} and describe their effects. |

