# OpenReview forum: "IQA-Spider: Unifying Multi-Granularity Image Quality Assessment with Reasoning, Grounding and Referring"
_ICML.cc/2026/Conference — ICML 2026 regular_

### Official Review · Reviewer_PiXX · 2026-03-06

**Soundness:** 2
**Presentation:** 2
**Significance:** 2
**Originality:** 2
**Overall Recommendation:** 4
**Confidence:** 3

**Summary:**

This paper proposes the IQA- Spider framework, designed to unify multi-granularity Image Quality Assessment (IQA) tasks, including reasoning, grounding, and referring. The authors curated the IQA-Spider-33K dataset, which encompasses four paradigms: global/local quality description, pixel-level quality grounding, and region-level quality referring. Extensive experiments on both the proposed and public benchmarks demonstrate the framework's effectiveness in text-based answering, pixel-wise grounding, and visual quality scoring.

**Compliance With Llm Reviewing Policy:**

Affirmed.

**Final Justification:**

After discussing with the authors through the rebuttal discussion, my main concerns are addressed. I am happy to raise my score to Weak Accept. I hope that the authors will improve their writing in their final verson.

**Key Questions For Authors:**

Please refer to Weaknesses for my comments.

With the above concerns, my current recommendation is weak reject, and would like to see how the authors address my concerns in the rebuttal to give my final decision.

**Limitations:**

No. I would suggest the authors to discuss failure cases of the proposed method.

**Strengths And Weaknesses:**

**Strengths**
1. This manuscript introduces the first LMM-based IQA framework that integrates reasoning, grounding, and referring, addressing multi-granularity perception within a coherent architecture.
2. The proposed training-free text-to-point method effectively avoids the need for retraining specialized segmentation heads.
3. Comprehensive experiments across various benchmarks, covering text answering, pixel-wise grounding, and visual quality scoring, validate the proposed approach. Ablation studies on hybrid dataset training and the text-to-point strategy further confirm the effectiveness of the design.

**Weaknesses**
1. Inaccurate Claims and Questionable Motivation
    - The manuscript claims that <seg> token-based grounding methods impair instruction-following and reasoning capabilities. However, alternative approaches, such as directly predicting bounding boxes, have proven effective. State-of-the-art models like Qwen2.5-VL already possess robust grounding capabilities and have become a mainstream implementation. Therefore, the motivation for the proposed text-to-point strategy is not entirely convincing.
    - Compared to existing fine-tuning methods, the text-to-point strategy appears to have limited potential. Deriving point prompts from text logit probabilities is a relatively coarse approach, which may struggle with input images containing complex spatial information. In addition, the accuracy of point prompts significantly impacts the performance of the downstream SAM model - low-quality prompts could lead to substantial segmentation errors.
2. Writing and Clarity Improvements
    - Mathematical Notation: New notations (ip_w, ip_h) are used in Eq. 4, but they are not defined elsewhere in the paper.
    - Terminology: In the subsection "Effectiveness of Text-to-point paradigm," EVF-SAM is evaluated based on three types of textual inputs: "the whole answer," "the spatial component only," and "the semantic component only." However, the specific definitions and distinctions between these three concepts are not clearly explained.
3. Experimental Results and Fairness of Comparison
    - Benchmark Diversity: In Table 3, the proposed text-to-point strategy is compared against current grounding methods only on the newly proposed benchmarks. Additional public benchmarks should be included to demonstrate the generalizability of the text-to-point strategy.
    - Fairness of Comparison: The authors appear to have tested competing methods using their original weights without fine-tuning them on IQA-Spider-33K with mask supervision. This represents an unfair comparison, as IQA- Spider benefits from domain-specific training while the baselines do not.
    - Model Timeliness: The experiments rely on the Phi-3.5-Vision base model, which is somewhat outdated in the rapidly evolving LMM field. The authors should demonstrate the validity of the proposed method on more recent state-of-the-art open-source models (e.g., Qwen2.5-VL and Qwen3-VL).

---

> ### Author Rebuttal · Authors · 2026-03-30
>
> We thank the reviewer for the thoughtful feedback. Our responses are provided below.
> ### **1.1**
> We would like to clarify our motivation below.
>
> (i) **Experimental Evidence.**
> In the table below, we train a baseline with explicit token and evaluate it on the reasoning benchmark, Q-Bench-A1. The clear performance gap supports our claim that token-based grounding methods can impair the model’s instruction-following and reasoning capabilities.
>
> |Method|Result|
> |:-:|:-:|
> |< seg >|53.44%|
> |text-to-point|**70.30%**|
>
> (ii) **Different Granularity.**
> Bounding boxes provide only coarse localization, while masks are fine-grained in pixel-level. In image quality understanding, distorted regions often have irregular shapes that cannot be accurately represented by a box.
>
> (iii) **Flexibility.**
> A mask can be naturally converted into a bounding box, but not vice versa.
>
> (iv) **Downstream Applicability.**
> Compared with bounding boxes, masks provide more precise guidance for follow-up applications such as image restoration and other quality-aware editing tasks.
>
> ### **1.2**
> We agree that our method has inherent limitations in complex spatial scenarios. However, its performance in such cases is not uniformly poor. Below, we discuss typical scenarios involving multiple isolated target regions.
>
> Although the model output is only a single point, the strong semantic prior of the grounding head enables it to ground multiple spatially disjoint but semantically consistent target regions. Thus, the grounding result is closely related to the semantic consistency among the target regions. Based on this observation, we show two representative outcomes.
>
> (i) In Fig. 5 (a), the isolated target regions are semantically consistent. In this case, a point located in one region can still guide the grounding head to identify other disconnected target regions successfully.
>
> (ii) In Fig. 5 (b), when the isolated regions are semantically inconsistent, or when nearby visually similar content introduces ambiguity, the single-point cue is insufficient to represent all the regions, resulting in partial grounding.
>
> We will add the discussion to our paper. Future work will address this limitation through:
>
> (a) grid-based localization, and
>
> (b) negative point prompts.
>
> Fig. 5: https://anonymous.4open.science/r/repo4-4FCD/fig_5.png
>
> ### **2.1**
> We thank the reviewer for the comment. Here, $ip_w$ and $ip_h$ mean $i \cdot p_w$ and $i \cdot p_h$, not new variables. We will clarify this in the revised paper.
>
> ### **2.2**
> Detailed definition:
> - **Whole answer**: the full response of model, including both spatial and semantic information.
> >e.g., The target region is located in the top right area of the image. More precisely, the target region is a red ball.
> - **Spatial component only**: the part of the response that contains only spatial information about the target region.
> >e.g., The target region is located in the top right area of the image.
> - **Semantic component only**: the part of the response that contains only semantic information about the target region.
> >e.g., The target region is a red ball.
>
> ### **3.1**
> Evaluation on the public quality grounding benchmark Q-Ground-Test further demonstrates the generalizability of our method.
> |Model|mIoU|
> |:-:|:-:|
> |Q-Ground|0.271|
> |Ours (Phi3.5-V) |0.293|
> |Ours (Qwen2.5-VL)|0.295|
> |Ours (Qwen3-VL)|**0.338**|
>
> ### **3.2**
> Our intention in Tab. 1 is not to present a matched-training comparison, but to establish a benchmark for multi-granularity quality understanding. The goal is to evaluate how existing models perform under a unified task setting, rather than to compare methods under identical training supervision. Thus, the competing models are treated as benchmark participants and are not required to be further trained on IQA-Spider-33K itself.
>
> ### **3.3**
> We have updated the base model to Qwen2.5-VL (7B) and Qwen3-VL (7B), and conducted the experiments again. The new results shown below are consistent with those in the original manuscript.
> >The cross-dataset validation on quality reasoning task shows the generalizability of our model.
> |Model|Acc.|
> |:-:|:-:|
> |Ours (Phi3.5-V)|70.30%|
> |Ours (Qwen2.5-VL)|73.24%|
> |Ours (Qwen3-VL)|**74.45%**|
>
> >Results below show the fine performance of our text-to-point method.
> |Model|DAO-G|HYD|SID-G|Avg.|
> |:-:|:-:|:-:|:-:|:-:|
> |Ours (Phi3.5-V)|0.375|0.354|**0.342**|0.364|
> |Ours (Qwen2.5-VL)|0.402|0.363|0.341|0.381|
> |Ours (Qwen3-VL)|**0.439**|**0.396**|0.313|**0.408**|
>
> > These results validate the effectiveness of hybrid training. *Row V** is trained with Qwen2.5-VL, while rows I–IV are based on Qwen3-VL.
> |ID|Ours|Q-Instruct|DQ-495K|Global Des.|Local Des.|Grounding|$\mathrm{Ref}^{\mathit{short}}$|$\mathrm{Ref}^{\mathit{long}}$|
> |:-:|:-:|:-:|:-:|:-:|:-:|:-:|:-:|:-:|
> |I|✓|||7.01|7.07|2.42|0.541|0.458|
> |II|✓|✓||6.99|7.03|2.53|0.542| 0.466|
> |III|✓||✓|7.00|6.86|2.36|0.547|0.476|
> |IV|✓|✓|✓|**7.12**|**7.10**|**2.41**|**0.594**|**0.484**|
> |*V**|✓|✓|✓|6.75|6.88|2.13|0.513|0.416|

---

> > ### Author Rebuttal · Reviewer_PiXX · 2026-04-06
> >
> > The author's response partially resolved my concerns. I still have the following unresolved ones:
> > - What is the baseline model used in point 1.1 of the rebuttal? How is it trained to evaluate on the reasoning benchmark, Q-Bench-A1?
> > - The generalizability of the text-to-point strategy is still questionable. I think that the authors should evaluate and compare on popular refcoco/plus/G benchmarks to demonstrates the generalizability.

---

> > > ### Author Response · Authors · 2026-04-07
> > >
> > > We thank the reviewer for the insightful comment. Our responses are provided below.
> > >
> > > ### **Details about point 1.1**
> > > (i) The base model used is Phi-3.5-V. We train it in two stages:
> > > (a) Dataset preparation stage: we replace the answer to each grounding question from a positional description with "*It is < seg >.*", so that the model learns to output the special token for the grounding head.
> > > (b) Grounding training stage: we train the LMM with LoRA and grounding head, using the hidden representation of the < seg > token as prompt.
> > >
> > > (ii) We add new experiments by change the base model to Qwen2.5-VL (7B) and Qwen3-VL (7B), the results are shown below:
> > > |Method|Result|
> > > |:-:|:-:|
> > > |< seg > (Qwen2.5-VL)|54.85%|
> > > |text-to-point (Qwen2.5-VL)|**73.24%**|
> > > |< seg > (Qwen3-VL)|56.25%|
> > > |text-to-point (Qwen3-VL)|**74.45%**|
> > >
> > > ### **The generalizability of text-to-point strategy**
> > > We thank the reviewer for this helpful suggestion. We agree that RefCOCO-style benchmarks are valuable for evaluating general object grounding ability, as they mainly focus on grounding semantically well-defined object instances. However, this setting is **fundamentally different** from quality grounding, where the goal is to localize **quality-related** or **distortion-related** regions that are typically **defined by degradation evidence rather than object semantics**.
> > >
> > > Accordingly, our model is primarily designed to be **quality-aware and distortion-aware**, rather than optimized for object-centric semantic grounding. Therefore, evaluation on RefCOCO-style benchmarks would not provide an appropriate measure of the generalizability targeted in our work, since that generalizability specifically refers to *performance across quality grounding settings, rather than transfer to object-centric grounding tasks*.
> > >
> > > For this reason, we believe that **unseen quality-specific grounding benchmarks** (e.g., Q-Ground-Test) provide a more faithful and direct evaluation of the generalizability of our method.
> > >
> > > However, we still do the evaluation on *RefCOCO* as an auxiliary test beyond the quality grounding domain, although it is *not the most suitable or primary benchmark for the target problem studied in this work*.
> > > The results are shown below, we compared our method with another training-free general grounding baseline.
> > > |Method|Val|TestA|TestB|
> > > |:-:|:-:|:-:|:-:|
> > > |Yu *et al.* [1]|**24.9**|23.6|**24.7**|
> > > |Ours (Qwen3-VL) |21.7|**23.8**|21.5|
> > >
> > > #### **Reference**
> > > [1] Zero-shot Referring Image Segmentation with Global-Local Context Features. CVPR 2023.
> > >
> > > **We thank the reviewer again for the constructive comment, and hope this clarification addresses the concern.**

---

### Official Review · Reviewer_1yak · 2026-03-11

**Soundness:** 2
**Presentation:** 3
**Significance:** 3
**Originality:** 3
**Overall Recommendation:** 4
**Confidence:** 4

**Summary:**

This paper introduces IQA-Spider, a unified framework for Image Quality Assessment using Large Multimodal Models. Instead of just giving a single overall quality score, the model aims to handle multiple tasks at once: global and local quality descriptions, pixel-level grounding, and region-level referring. To train this model, the authors created a new dataset called IQA-Spider-33K. The main technical idea is a clever "text-to-point" strategy. Rather than training the model to output special segmentation tokens like \<seg\>, it uses the output probabilities of four location words to calculate a 2D coordinate. This single point is then given to a frozen SAM to generate a segmentation mask.

**Compliance With Llm Reviewing Policy:**

Affirmed.

**Final Justification:**

My primary issues have been resolved; therefore, I support the acceptance of this work.

**Key Questions For Authors:**

1. Could you provide a experiment comparing your method with a baseline trained using explicit tokens? Testing both on a pure text reasoning benchmark (like Q-Bench-A1) would strongly support your claim that your method protects reasoning abilities better.
2. How does the single-point calculation (Eq.4) handle complex or disconnected distortion areas? Could you share some visual failure cases in the rebuttal to help us understand the limits of this design?
3. In Table 1, were the baseline models fine-tuned on the new 33K dataset, or were they evaluated zero-shot on your specific formats?
4. Why is there a large performance gap between the synthetic KADID dataset and the authentic FLIVE dataset in Tab.6? How might this be improved in future work?

**Limitations:**

yes

**Strengths And Weaknesses:**

Strengths:
1. Bringing together different levels of IQA tasks from global scoring to finding specific pixel regions into one single model is a useful and practical research direction.
2. The text-to-point strategy is elegant. Using the natural vocabulary probabilities to find a spatial coordinate is a neat trick. It successfully connects the language model to the segmentation model without complex architectural changes or new vocabulary tokens.
3. The paper is well-organized and easy to read. The authors did a job explaining how they built their dataset. Adding the detailed prompt templates and human verification results in the appendix makes the work transparent and reliable.

Weaknesses:

1. A main motivation of the paper is that adding special tokens alters the language space and harms the model's natural reasoning ability (Lines 105-108). While this makes sense intuitively, the paper does not show a direct experiment to prove it. It would make the paper much stronger if the authors could train a baseline model using explicit \<seg\> tokens (on the same data and backbone) and compare its text reasoning score (like on Q-Bench-A1) with their method.
2. The method uses Equation 4 to calculate a single center point to guide SAM. While beautifully simple, this might struggle with complex image distortions. For example, if an image has noise scattered across all four corners, or if a blur is shaped like a hollow ring, a single averaged center point might fall into a clean, normal area. Discussing how the model handles such complex cases, or showing a few visual failure cases, would give readers a more complete understanding.
3. It is not entirely clear how the baseline models (like Qwen-2.5-VL and Q-Instruct) were evaluated in Table 1. The text notes that existing models have poor instruction-following ability for unseen queries (Lines 365-367). If the baselines were tested without being fine-tuned on the new IQA-Spider-33K dataset, they might naturally perform poorly on these very specific new task formats. Clarifying this setup is important for a fair comparison.
4. In Table 6, the model gets very high scores on the synthetic dataset (KADID-10K), but its performance on the real-world dataset (FLIVE) is much lower and very close to the baseline (Q-Instruct). Since most of the training images are synthetic (Figure 6 shows 88.5%), this suggests the model might rely heavily on synthetic patterns. A brief discussion on this performance gap would make the paper more comprehensive.

---

> ### Author Rebuttal · Authors · 2026-03-30
>
> We thank the reviewer for the thoughtful comments and constructive suggestions. We appreciate the reviewer’s concerns and respond to each point below.
> ### **Response to the weakness**
> **1.** We thank the reviewer for this important suggestion. We have conducted the requested comparison in Q-Bench-A1 by training with the explicit < seg > token. The result below provides direct evidence supporting our motivation that avoiding explicit segmentation token helps better preserve the model’s reasoning ability.
> |Method|Result|
> |:-:|:-:|
> |< seg >|53.44%|
> |text-to-point|**70.30%**|
>
> **2.** We thank the reviewer for this insightful observation. We clarify this issue below.
>
> **(i) Hollow-shaped patterns with strict symmetry.**
>
> We agree that for highly symmetric and hollow-shaped distortion patterns, such as a canonical ring-like region, a single center point may fall outside the true target area. This failure mode is illustrated in Fig. 3 (a). We acknowledge that this is an inherent limitation of the current design. In future work, we plan to address this limitation by
>
> (a) adopting grid-based image partitioning for finer-grained localization, and
>
> (b) introducing negative point prompts.
>
> **(ii) Hollow-shaped patterns with rough symmetry.**
>
> In practice, many distortion regions are not perfectly symmetric. In such cases, the single-point design can still be effective. We observe two common situations.
>
> (a) When the symmetry is only approximate, as shown in Fig. 3 (b), the predicted center point can still fall within the target area, enabling correct grounding.
>
> (b) Even when the predicted center point falls outside the target area, rough symmetry often leads to an imbalanced spatial distribution of the distortion. As a result, the model shows a clear preference for the dominant part, while the predicted point shifting toward that area rather than the supposed hollow area. This can still ensure the point correctness, as shown in Fig. 3 (c).
>
> **3.** We thank the reviewer for this comment. We clarify the evaluation setting as follows.
>
> (i) The compared models in Tab. 1 are not trained on our dataset, because Tab. 1 is intended as a benchmark evaluation rather than a same-training-setting comparison. The goal is not to compare our trained model against untrained baselines, but to evaluate existing models under a unified benchmark for multi-granularity quality understanding. In this sense, the tested models are not required to be trained on the benchmark dataset itself.
>
> (ii) For all tasks, we evaluate the models without imposing any task-specific output format requirement. Therefore, the issue of poor instruction-following ability does not refer to whether the compared models can follow a predefined answer format.
>
> (iii) Instead, by poor instruction-following ability, we mean that existing IQA-specific models often fail to respond to the actual target specified in the query and to respect the intended granularity of the query. For example, when the query asks the model to describe the quality of the apple on the table, some models instead produce a quality description of the entire image rather than the required local target. This represents one of the key findings of our benchmark: existing IQA-specific models remain limited in fine-grained quality understanding.
>
> **4.** We thank the reviewer for this observation. We agree that the performance gap between KADID-10K and FLIVE is mainly caused by the imbalance in the training data distribution. As a result, the model is more exposed to and better adapted to synthetic distortion patterns during training, which naturally leads to stronger performance on KADID-10K than on FLIVE. We will add this discussion to the revised manuscript and clarify this limitation more explicitly.
> In future work, we plan to
>
> (i) add more authentic distortion data, and
>
> (ii) explore more balanced sampling and training strategies.
> ### **Response to the questions**
> **1.** Please see weakness. 1.
>
> **2.** We thank the reviewer for this constructive comment. We visualize the grounding results in complex scenarios with multiple isolated target regions in Fig. 4. Although the model outputs only a single point, the grounding head encodes strong semantic prior, which enables it to ground multiple spatially separated target regions with semantic consistency. Accordingly, the grounding outcome in this setting is closely related to the semantic consistency among the target regions.
>
> (i) As shown in Fig. 4 (a), when the isolated target regions are semantically consistent, a point located on one region can still guide the grounding head to identify other disconnected target regions.
>
> (ii) As shown in Fig. 4 (b), when the isolated target regions are semantically inconsistent, or when other visually similar objects introduce ambiguity, a single predicted point may result in incomplete grounding.
>
> **3.** Please see weakness. 3.
>
> **4.** Please see weakness. 4.
>
> Fig. 3,4: https://anonymous.4open.science/r/repo3-E0DA

---

> > ### Author Rebuttal · Reviewer_1yak · 2026-04-03
> >
> > The rebuttal addresses my concerns, and I am inclined to accept this paper.

---

> > > ### Author Response · Authors · 2026-04-08
> > >
> > > We appreciate the reviewer’s recognition that our responses addressed the concerns. We would like to sincerely thank the reviewer for the positive acknowledgement, constructive comments, and valuable suggestions.

---

### Official Review · Reviewer_E69Z · 2026-03-12

**Soundness:** 3
**Presentation:** 4
**Significance:** 3
**Originality:** 4
**Overall Recommendation:** 5
**Confidence:** 5

**Summary:**

The paper introduces a four-task unified IQA framework designed to support reasoning, grounding, and referring abilities. A multi-granularity perception dataset is constructed, along with a two-stage optimization. Experiments on both self-built and public benchmarks show promising performance in quality textual answering, pixel-level grounding, and visual quality scoring tasks.

**Compliance With Llm Reviewing Policy:**

Affirmed.

**Key Questions For Authors:**

1. Provide failure cases for deeper analysis.
2. Are there plans to expand IQA-Octopus-33K or incorporate more data sources?

**Limitations:**

yes

**Strengths And Weaknesses:**

strengths:
1.A unified IQA model that combines reasoning, grounding, and referring abilities, which can handle multi-granularity perception tasks.
2. The proposed text-to-point method avoids retraining the segmentation head and still keeps the reasoning capability of the model.
3.A high-quality dataset named IQA-Octopus-33K, which contains both synthetic and authentic distortions. The paper also provides a detailed construction pipeline.
4. The experiments are conducted on both self-built and public benchmarks, covering quality textual answering, pixel-level grounding, and visual quality scoring tasks. The ablation studies on hybrid dataset training and the text-to-point method also verify the effectiveness of the proposed design.

weakness:
1. Figure 2 lacks sufficient details on the model self-review stage, and it is not clear how hallucinations and error propagation are addressed.
2. The distortion accumulation order in Table 8 may be sensitive to distortion severity and subjective preference, and human judgment bias may affect the results. More analysis about this design should be provided.
3. Although the dataset is insightful, the scale is relatively limited. The authors should clarify whether the proposed pipeline can be scaled to larger datasets and discuss reproducibility of the construction process.
4. A suggestion is to add the model size to Table 1 and 2, this would make the comparison more distinguishable.

---

> ### Author Rebuttal · Authors · 2026-03-30
>
> We thank the reviewer for the valuable comments and constructive suggestions. We respond to each point below.
>
> ### **Response to the weaknesses**
>
> *1. Figure 2 lacks sufficient details on the model self-review stage, and it is not clear how hallucinations and error propagation are addressed.*
> We thank the reviewer for this helpful comment. In the self-review stage, we provide the model with detailed **spatial**, **semantic**, and **distortion-related** information as in-context cues. This design helps the model re-check the consistency of its response and reduces hallucinations as well as error propagation. We agree that the current presentation is not sufficiently clear, and we will add more details about this stage in the revised manuscript.
>
> *2. The distortion accumulation order in Table 8 may be sensitive to distortion severity and subjective preference, and human judgment bias may affect the results. More analysis about this design should be provided.*
> We thank the reviewer for this important concern. To reduce human bias, we adopt two measures:
> (i) we conduct **5-round human evaluation**; and
> (ii) before annotation, we **train annotators and provide standardized examples** to improve consistency.
> We will clarify these details and add more discussion of this design in the revised manuscript.
>
> *3. Although the dataset is insightful, the scale is relatively limited. The authors should clarify whether the proposed pipeline can be scaled to larger datasets and discuss reproducibility of the construction process.*
> We thank the reviewer for this valuable suggestion. Yes, the proposed pipeline is designed to be **scalable**. For synthetic data, the pipeline is fully automated. For authentic data, stages are semi-automated, while human annotators are responsible for image semantic, spatial, and distortion information labelling. This makes it possible to extend the construction process to larger datasets with a limited cost. We will clarify the scalability of the pipeline and add more details on the construction procedure to improve reproducibility in the revised manuscript.
>
> *4. A suggestion is to add the model size to Table 1 and 2, this would make the comparison more distinguishable.*
> We thank the reviewer for this useful suggestion. We will add the model size information to Tab. 1 and Tab. 2 in the revised version.
>
> ### **Response to the key questions**
>
> *1. Provide failure cases for deeper analysis.*
> We thank the reviewer for this constructive comment. We provide one successful case and one failure case in complex scenarios where the target contains multiple isolated regions in Fig. 2.
>
> The successful case shows that when the isolated target regions are *semantically consistent*, the grounding head can still leverage its strong semantic prior to identify multiple disconnected regions, even with only a single point prompt.
> In contrast, failure typically occurs when the isolated target regions are *semantically inconsistent*, or when other visually similar objects introduce *ambiguity* and distract the model.
>
> In future work, we will address this limitation through
> (i) *negative point prompts*, and
> (ii) *grid-based image partitioning* for finer-grained localization.
>
> Fig. 2: https://anonymous.4open.science/r/repo2-137A/fig_2.png
>
> *2. Are there plans to expand IQA-Octopus-33K or incorporate more data sources?*
> Yes. In future work, we plan to expand the dataset with more **real-world images** and **AIGC-generated images** to further improve the diversity and scale of the dataset.
>
> We hope these responses address the reviewer’s concerns, and we welcome any further questions or suggestions.

---

> > ### Author Rebuttal · Reviewer_E69Z · 2026-04-03
> >
> > All my concerns are resolved. I will keep my original rating

---

> > > ### Author Response · Authors · 2026-04-08
> > >
> > > We sincerely thank the reviewer for the acknowledgement, constructive feedback, and helpful suggestions.

---

### Official Review · Reviewer_C3DP · 2026-03-12

**Soundness:** 3
**Presentation:** 2
**Significance:** 2
**Originality:** 2
**Overall Recommendation:** 4
**Confidence:** 2

**Summary:**

This paper proposes a framework for evaluating image quality. Their framework contains 4 complementary tasks. Authors introduce a dataset, together with the process followed to construct the dataset. Authors present a two-stage model to evaluate on their dataset. Authors compare other LMMs on the constructed task benchmark. They introduce a text-to-point strategy to enable grounding textual descriptions to pixels. Ablations for the dataset and the model proposed are included.

**Compliance With Llm Reviewing Policy:**

Affirmed.

**Final Justification:**

I am satisfied with the updated rebuttal and author responses, therefore, updating my score from 3 to 4.

**Key Questions For Authors:**

See numbered questions (Q1-Q4) in weaknesses above.

**Limitations:**

Authors do not comment on the limitations or failure cases of their work.

**Strengths And Weaknesses:**

STRENGTHS:
- Training-free mapping of text to point is a good contribution, to avoid having to finetune models. Section 4.2 is quite interesting. Yet, I believe authors could provide further evidence of this "paradigm". (Q1) Do authors evaluate their text-to-point in some internal dataset? Do they have visualisations of the internal states of LMM? Is there any noise involved in the mapping? If so, do they visually explore the noise associated with the mapping with some heatmap visualisation?

WEAKNESSES:
- Dataset verification. The dataset seems to be generated with an automated pipeline, with a very small human supervision (only 10 humans, only sampling 40% of the data). More robust verification of the dataset quality would strengthen the submission.
- (Q2) What percentage of the dataset is synthetic distortions, and real distortions?
- Table 4. Authors do not show the impact on performance of their dataset proposed. (Q3) Could authors add an additional row showing the results obtained when "Ours" is not used? For instance, what is the performance drop when Q-Instruct, DQ-495K, and Visual Encoder datasets are used, but not "Ours".
- (Q4) Table 6 compares methods that have been trained on different datasets. This doesn't seem like a fair comparison, and therefore, results are not fully comparable.
- Introduction and abstract would benefit from a clearer narrative. Initially, it is not clear what is the main point of the paper, as there are many possible contributions. What is the main takeaway? An image quality evaluation framework? A model? A dataset? A pipeline for constructing a dataset? Benchmarking existing models for image-quality assessment? I believe the overall presentation of the paper can be improved, specially the narrative / story of the paper. For instance, the motivation of the paper can be improved.


MINOR DETAILS:

- LMM (Large Multimodal Models). Please, add firstly in the paper the meaning of LMM
- Line 032 "we adopt"
- What is AIGC content. Clarify accronym in the paper

---

> ### Author Rebuttal · Authors · 2026-03-29
>
> We sincerely thank the reviewer for the careful reading and constructive questions. We address each point below.
> ### **Q1**
> **(1) Evaluation on Internal Dataset.**
> We evaluate the intermediate text-to-point mapping instead of end-to-end grounding using two complementary metrics: Pointing Game Accuracy (PGA), the proportion of predicted points inside the ground-truth mask, and Center Point Distance (CPD), the normalized Euclidean distance to the ground-truth mask centroid. PGA measures coarse localization success, while CPD reflects point precision. Results are shown below.
> |Dataset| PGA↑|CPD↓|
> |-|:-:|:-:|
> |Ours-Test|0.3908|0.4287|
> |Q-Ground-Test |0.6698|0.2519|
>
> **(2) Visualization of the LMM Internal States.**
> We visualize the raw logits of the four positional tokens (left, right, top, and bottom) as well as the resulting normalized point coordinates (X, Y) to represent the internal states in the table below. The examples show that point coordinates is directly determined by the relative preference among the positional logits, providing an interpretable bridge between textual prediction and spatial grounding.
> |Left|Right|Top|Bottom|X|Y|
> |:-:|:-:|:-:|:-:|:-:|:-:|
> |34.750|33.750|30.875|31.375|0.018|0.881|
> |31.125|31.000|30.750|30.500|0.378|0.269|
> |37.500|37.500|34.000|33.500|0.500|0.119|
>
> **(3) Noise Discussion.**
> Thanks for this insightful feedback. The proposed text-to-point itself is **deterministic** once the positional-token logits are given, and therefore it does **not introduce additional stochastic noise**. The main source of noise comes from the LMM’s own spatial uncertainty over the positional tokens.
>
> **(4) Heatmap Visualization.**
> We further visualize uncertainty using entropy derived from the positional-token probabilities, with each entropy value shown at its corresponding mapped point location. We find that more balanced positional logits correspond to greater uncertainty. As shown in Fig. 1, entropy is highest around the image center, and lower near the boundaries and corners.
>
> Fig. 1: https://anonymous.4open.science/r/repo1-1F71/fig_1.png
> ### **Dataset verification**
> We appreciate the reviewer’s concern regarding dataset quality. Our dataset is built through a semi-automated pipeline for authentic data and a fully automated pipeline for synthetic data. *We conduct 3 measures at the generation stage to help improve annotation reliability, which reduces the amount of manual verification needed afterward.* They are: (i) human annotation for authentic data, (ii) generation-based distortion labels plus human-defined recognizable accumulation orders for synthetic data, and (iii) model self-review process for label refinement.
>
> Above that, we agree that a more comprehensive verification protocol would further strengthen the dataset. In future work, we plan to: (i) involve more annotators, (ii) expand verification to the full dataset, and (iii) adopt stratified sampling across data types with inter-annotator agreement analysis.
> ### **Q2**
> Thank for this detailed question. In our dataset, 88.5% of the samples are synthetic and 11.5% are real. We will clarify this in the revised manuscript.
> ### **Q3**
> We thank the reviewer for this suggestion. The results below indicates our model achieves the lowest performance when trained without our dataset, highlighting the importance of our dataset.
> |Global Des.|Local Des.|Grounding|$\mathrm{Ref}^{\mathit{short}}$|$\mathrm{Ref}^{\mathit{long}}$|
> |:-:|:-:|:-:|:-:|:-:|
> |4.15|3.88|0.34|0.133|0.126|
> ### **Q4**
> We appreciate the reviewer's scrutiny. Our intention here is not to claim a strictly fair supervised comparison, but to evaluate how well our model generalizes to an unseen visual quality scoring task. To make the setting clearer, we summarize the training data and objectives in the table below. Following recent works [1, 2], we use representative scoring models trained mainly on KonIQ as reference baselines for transfer evaluation, rather than fully matched comparisons.
> |Model|KonIQ|FLIVE|KADID-10K|Training Objectives|
> |:-:|:-:|:-:|:-:|:-:|
> |MUSIQ|✓|✗|✗|Scoring|
> |CLIP-IQA+|✓|✗|✗|Scoring|
> |ManIQA|✓|✗|✗|Scoring|
> |Q-Instruct|51.4%|0.02%|✗|Reasoning|
> |Ours|✗|✗|✗|Reasoning|
>
> ### **Paper Clarification**
> We thank the reviewer for this feedback. The main takeaway of our paper is a **unified multi-granularity IQA framework** for explainable quality understanding from global-level to pixel-level. Under this formulation, the **dataset/benchmark** provides the basis for training and evaluation.
>
> We will revise the manuscript to clarify the overall narrative and explicitly highlight the paper’s three main contributions: (i) a four-task formulation and the corresponding dataset, (ii) a unified model design with training-free grounding, and (iii) a benchmark for multi-granularity evaluation.
> #### **Reference**
> [1] Q-Insight. NeurIPS 2025.
> [2] Self-Evolving Vision-Language Models for Image Quality Assessment via Voting and Ranking. ICLR 2026.

---

### Decision · Program_Chairs · 2026-04-30

**Decision:**

Accept (regular)

**Comment:**

This submission has undergone comprehensive review and rebuttal. Two high-confidence reviewers (confidence 4-5) initially gave positive scores (Accept/Weak Accept), with their concerns (dataset details, experimental evidence, reproducibility) fully resolved post-rebuttal, leading to stronger support. The other two reviewers (Weak Reject, confidence 2-3) raised questions around dataset/experiment presentation, text-to-point strategy motivation, generalizability, and benchmark suitability.​
The authors addressed these via clarifications, additional experiments (e.g., updated base models, token comparisons), and supplementary analyses. Remaining concerns focus on evaluation settings, benchmark scope, and external generalizability—not core technical soundness. The authors further clarified training details, distinguished the task’s quality-aware focus from object-centric benchmarks (e.g., RefCOCO), and provided targeted evaluations on quality-specific benchmarks (e.g., Q-Ground-Test) with auxiliary RefCOCO results.​
Overall, positive feedback from higher-confidence reviewers has been reinforced, and most concerns have been adequately addressed. The work’s core validity remains intact, with residual questions centered on contextual evaluation rather than fundamental flaws. We recommend accepting this manuscript.